

# Is considering runs (in)consistency so useless for weather forecasting?

Hugo Marchal [1], François Bouttier [1], and Olivier Nuissier [1]

[1]CNRM, Toulouse University, Météo-France and CNRS, Toulouse, France

**Correspondence:** Hugo Marchal (hugo.marchal@meteo.fr)

**Abstract.** This paper addresses the issue of forecasting the weather, using consecutive runs of one given numerical weather prediction (NWP) system. In the literature, considering how forecasts evolve from one run to another has never been proved relevant to predicting the upcoming weather. That is why the usual approach to deal with this consists of blending all together the successive runs, which leads to the well-known "lagged" ensemble. However, some aspects of this approach are questionable, and if the relationship between changes in forecasts and predictability has so far been considered weak, this does not mean that the door is closed. In this article, we intend to further explore this relationship by focusing on a particular aspect of ensemble prediction systems, the persistence of a given weather scenario over consecutive runs. The idea is that, the more it persists over successive runs, the more it is likely to occur, but its likelihood is not necessarily estimated as it should be by the latest run alone. Using the regional ensemble of Météo-France, AROME-EPS, and forecasting the probability of certain (warning) precipitation amounts being exceeded in 24 hours, it has been found that reliability, an important aspect of probabilistic forecast, is highly sensitive to that persistence. The present study also shows that this dependency can be exploited to improve reliability, for both lagged ensembles and individual runs. From these results, some recommendations for forecasters are made, and the use of new predictors for statistical post-processing, based on consecutive runs, are encouraged. The reason for such sensitivity is also discussed, leading to a new insight on weather forecasting using consecutive ensemble runs.

## 1 Introduction

Meteorological centers have much improved their Numerical Weather Prediction (NWP) systems over the past years, as summarized in Bauer et al. (2015). These improvements come in many forms and shapes, and some of them consist of major changes, increasing for example the model resolution (ECMWF with IFS (ECMWF, 2016, 2023), DWD with ICON (Deutscher Wetterdienst, 2022), Météo-France with AROME-France (Brousseau et al., 2016)), the ensemble size (Environment Canada with their global ensemble (Charron et al., 2010); NCEP with GEFS (Zhou et al., 2022)), or even the frequency with which forecasts are refreshed, i.e. the number of runs per day for a given model.



This last strategy is well illustrated by the Met Office, their regional ensemble MOGREPS-UK being run four times a day
up to March 2019, and every hour since then (although less members are being produced at each run, cf Porson et al., 2020).
Moreover, their global ensemble, MOGREPS-G, has gone from two runs per day to four (Hagelin et al., 2017). Other centers
have also adopted this strategy, such as Météo-France whose AROME-France, their regional deterministic model which was
initially run four times a day (Seity et al., 2011), is now run every three hours since July 2022. Likewise, AROME-EPS, its
ensemble version, is run four times a day since March 2018 compared to two previously (Raynaud and Bouttier, 2017).

In our opinion, increasing run frequency significantly affects how the weather is forecast. Along with the extension of
forecast range, it leads to the overlapping in time of consecutive runs, so that nowadays, a given NWP forecast is usually
considered by forecasters in combination with previous ones, especially if they are close in time. In this context, the variations
from one run to another inevitably become an important information to deal with, in particular for decision making. However,
this aspect of weather forecasting is relatively unexplored in the literature, as pointed out by Ehret (2010) or more recently
Richardson et al. (2020).

A review of the literature shows that most related studies have been concerned with quantifying run-to-run variability,
underlining some features such as trends, convergence, consistency, or on the contrary "jumpiness". Considering deterministic
models as well as ensembles, several measures have been proposed and tested on various parameters, including geopotential
height (Zsoter et al., 2009), sea level pressure (Hamill, 2003), temperature (Hamill, 2003; Zsoter et al., 2009; Griffiths et al.,
2019), rainfall (Ehret, 2010; Griffiths et al., 2019), wind direction (Griffiths et al., 2021), large-scale flow over the European-
Atlantic region (Richardson et al., 2020), or even tropical cyclone tracks (Fowler et al., 2015; Richardson et al., 2024). The
impact of run-to-run variability has also been studied from the point of view of decision making, for instance how the incurred
expense of a given decision can be influenced by the way forecasts evolve over successive runs (McLay, 2011), to "decide
now or wait for the next forecast?" (Jewson et al., 2021), by way of communication challenges about forecast changes (Jewson
et al., 2022).

Although interesting, these studies tend to be rather limited when it comes to concretely predicting the weather using con-
secutive runs that may differ from each other. The decision making studies partly address this issue, but are conducted within a
simplified theoretical framework that does not reflect the complexity of real-world decision making (Jewson et al., 2021, 2022).
Conversely, most run-to-run variability measures have been introduced to identify features in the evolution of forecasts, or to
assess their consistency, but never as an additional information to improve weather forecasting. Actually, the relationship be-
tween changes in forecasts and the upcoming weather (that is, predictability) has rarely been studied as such, and only few
insights can be found sporadically (Persson and Strauss, 1995; Hamill, 2003; Zsoter et al., 2009; Ehret, 2010; Pappenberger
et al., 2011; Richardson et al., 2020, 2024). It is suggested that forecast jumpiness is more a matter of modeling than pre-
dictability, and that there is no strong correlation between run-to-run variability and forecast error. Consequently, the ECMWF
Forecast User Guide advises forecasters not to rely on how a given NWP system behaves from one run to another (Owens and
Hewson, 2018). The usual handling of successive runs is then quite straightforward: either consider the most recent one, or
blend them all together to create a "lagged" ensemble, as done for example by the Met Office within IMPROVER (Roberts





et al., 2023). However, both approaches suffer from shortcomings, and further studies are needed to confirm the low usefulness of considering the evolution of forecasts for weather forecasting.

Only using the latest run can be a risky strategy, because even if it is the most skillful on average (as forecast error increases
with forecast range), it can sometimes be worse than the previous runs and mislead forecasters. In particular, many high-impact case studies have cast doubt over this strategy by showing the importance of the previous runs: Hurricane Laura in August 2020 with the ECMWF ensemble (Richardson et al., 2024), flash-flood during 15-16 June 2010 in the south-east of France with ARPEGE-EPS (Nuissier et al., 2012), torrential rainfall on 10 May 2006 over the southern United Kingdom with UK4 (Mittermaier, 2007), catastrophic flood in Bavaria in August 2005 with GFS (Ehret, 2010), and many more.

Conversely, considering the sequence of successive runs is usually done by converting them into an ensemble, known as a lagged ensemble. If this approach can improve forecast skill, it also assumes in most studies (Hoffman and Kalnay, 1983; Lu et al., 2007; Mittermaier, 2007; Ben Bouallègue et al., 2013) that all runs are equally likely, as the few attempts to weight runs were unsuccessful (Ben Bouallègue et al., 2013; Raynaud et al., 2015). Therefore, the chronological order of runs is not accounted for, and no distinction is being made between, for instance, a sequence of four deterministic runs, the two earliest
forecasting light rainfall whereas the two latest forecasting intense rainfall, and the opposite. Yet this distinction seems crucial, at least from the forecasters' point of view. More generally, an ensemble composed of successive runs reflects, by definition, the response of a given NWP system to the recent changes in the atmosphere, processed through its data assimilation algorithm. On the contrary, a standard "Monte Carlo" ensemble only reflects its sensitivity to many sources of uncertainty such as modeling approximations or initial conditions (Leutbecher and Palmer, 2008). These two ensembles do not carry the same information,
and in this respect, distinguishing a sequence of consecutive runs from a standard ensemble seems appropriate.

As it has been previously reported, this idea is somewhat at odds with the current state of the art, since the evolution of forecasts has never been found to be strongly related to the upcoming weather. Nevertheless, we believe that further studies are needed to clarify that statement. Indeed, many results published in earlier work were based on run-to-run variability measures whose relevance may be questioned, as it has been done for instance by Di Muzio et al. (2019) for the "jumpiness" index
described in Zsoter et al. (2009). The same criticism can also be applied to the parameters on which these studies are based. What has been found for the temperature or the geopotential height does not necessarily apply to other parameters such as precipitation, which is characterized by a larger variability in both space and time (Ebert and McBride, 2000; Roberts, 2008). Finally, the run-to-run variability issue has mostly been addressed for low-impact weather (except in Richardson et al. (2020, 2024)), while the relationship that might exist between changes in forecasts and predictability is often pointed out by
forecasters during possible high-impact weather. Some related case studies have been documented, and can be found in Kreitz et al. (2020); Caumont et al. (2021) (same event) or in Plu et al. (2024).

For all these reasons, in this article we intend to further explore the predictive skill of considering the evolution of forecasts. A lagged ensemble is therefore used, but taking into account the chronological order of run. This is done by focusing on a particular aspect of ensembles, the persistence of a given weather scenario over consecutive runs, and by investigating
what that means in terms of predictability. Using the regional ensemble of Météo-France, AROME-EPS, and forecasting the probability of certain (warning) precipitation amounts being exceeded in 24 hours, we study how the skill of the latest run or the



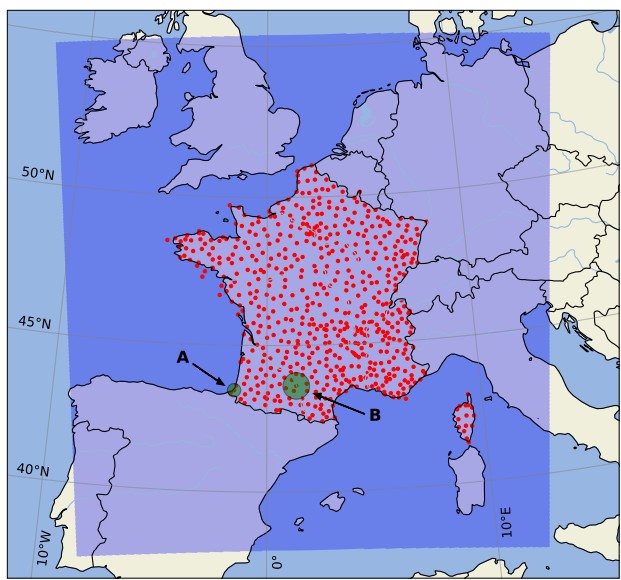

**Figure 1.** AROME-EPS domain is in shaded blue. The RADOME rain gauges are represented by red dots. The green disks represent 25-km and 50-km radius neighborhoods, respectively in the vicinity of Biarritz (A) and Toulouse-Blagnac (B).

standard lagged ensemble may vary, depending on the persistence of the targeted events over the successive runs. From these results, recommendations for forecasters are made, as well as general thoughts on the usefulness of considering the evolution of forecasts for weather forecasting.

This paper is organized as follows: section 2 introduces the data set, section 3 the methodology, section 4 the results, that are discussed in section 5 before the conclusion in section 6.

## 2 Data set

### 2.1 NWP system: AROME-EPS

AROME-EPS, the Météo-France high-resolution regional ensemble, is used in this study to assess the potential usefulness of considering run-to-run variability for weather forecasting. Being run four times a day since March 2018, at 03h00, 09h00, 15h00, 21h00 UTC, and making predictions up to 51h, AROME-EPS produces every day four forecasts coming from separate but close in time initializations, that overlap at least until the end of the following day, which makes it well suited to this study.

AROME-EPS comprises 17 members with 1.3km horizontal resolution and 90 vertical levels. One member is a control member, corresponding to AROME-France, the nonhydrostatic convection-permitting regional model of Météo-France (Seity et al., 2011; Brousseau et al., 2016). The other 16 perturbed members are obtained by sampling four sources of uncertainties: initial conditions (Raynaud and Bouttier, 2017), model errors (Bouttier et al., 2012), surface conditions (Bouttier et al., 2015), and lateral boundary conditions (Bouttier and Raynaud, 2018). For practical reasons, AROME-EPS fields are extracted on





a regular 0.025 x 0.025 degree latitude-longitude grid, using a simple nearest-neighbor algorithm. The domain covered by
AROME-EPS is displayed in fig 1.

## 2.2 Parameter of interest: 24-h accumulated precipitation

This paper focuses exclusively on 24-h accumulated precipitation. As mentioned in the introduction, accumulated precipitation
are characterized by large variability in both space and time, which makes this parameter challenging to predict, and likely to
vary significantly from one run to another as already experienced by many forecasters. The question of run-to-run variability
has also rarely been explored in terms of accumulated precipitation: only Ehret (2010) and Griffiths et al. (2019) did it, dealing
mostly with relatively light 3-h or 6-h rainfall accumulations. The 24-h accumulation period is preferred over shorter periods
especially because it summarizes what has fallen in a day, without considering how the accumulation is distributed over the 24
hours.

## 2.3 Observations and study period

The ANTILOPE quantitative precipitation estimate (QPE) algorithm (Champeaux et al., 2009) is used as 24-h accumulated
precipitation observations. ANTILOPE merges rain gauges data with radar reflectivity observations (Tabary, 2007). Because
ANTILOPE quality decreases with distance from radars and rain gauges, its use is restricted in this study to areas close to
rain gauges from the RADOME network. RADOME is the real-time meteorological observations network of Météo-France
(Tardieu and Leroy, 2003), and comprises 596 stations that are included by construction within the ANTILOPE analysis. Fig
1 shows the RADOME coverage over mainland France. How AROME-EPS forecasts will be compared to these observations,
and how ANTILOPE will be precisely used in this study will be detailed in the Methodology section.

Finally, the study period over which the results are obtained runs from early July 2022 to late June 2023, so approximately
one year of data. The day of 21 August 2022 has been removed due to missing AROME-EPS data.

## 3 Methodology

### 3.1 Use of the four daily AROME-EPS runs

This study focuses on daily precipitation, i.e. accumulated between 00h00 and 00h00 UTC. To predict the daily precipitation
on a given day, the four AROME-EPS runs of the day before are considered, the 21h00 UTC run being the latest one and
the 03h00 UTC run the oldest one, as depicted in fig 2. Note that hereafter, "Z21" stands for the 21h00 UTC run, "Z03" for
the 03h00 UTC run, and so on. Each run is used to predict the probability of occurrence of targeted events corresponding
to precipitation exceeding various thresholds during the following day. Such probabilities are computed using the frequentist
approach, which consists of counting the number of members which have simulated the exceedance, and dividing it by the total
number of members (17), assuming they are equally likely.





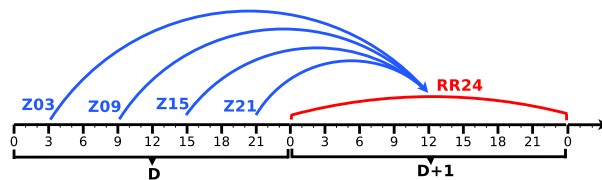

**Figure 2.** How the four daily AROME-EPS runs are used to forecast the 24-h accumulated precipitation ("RR24") of the following day.

## 3.2 The risk persistence and its diagnostic

The purpose of this study is to explore the possibility of using a sequence of successive runs other than through the standard lagging approach and to take advantage of it. To do so, our strategy is to exploit the chronological order of runs, which is usually not accounted for as explained in the introduction. Following discussions with Météo-France forecasters, we chose to focus on a particular aspect of ensembles, the persistence of a given weather scenario over consecutive runs. The idea is that, the more it persists over successive runs, the more it is likely to occur, but its likelihood is not necessarily estimated as it should be by the latest run alone. Here, the latest run is Z21, and the event for which probability is computed is daily precipitation exceeding a given threshold. In this respect, the variable "risk_persistence" is defined as the number of previous runs that predict a non-zero probability of occurrence, i.e that have at least one member simulating the exceedance. Hence, if by "risk" we mean "non-zero probability of occurrence":

$$
\text{risk\_persistence} = 
\begin{cases}
0 & \text{if none of Z03, Z09 and Z15 had predicted a risk} \\
1 & \text{if only one run among Z03, Z09 and Z15 had predicted a risk} \\
2 & \text{if only two runs among Z03, Z09 and Z15 had predicted a risk} \\
3 & \text{if Z03, Z09 and Z15 had all predicted a risk}
\end{cases}
\tag{1}
$$

The usefulness of risk persistence is assessed in two different but complementary ways, presented below. The relevance of its definition will be discussed after the results.

## 3.3 Manual assessment of the risk persistence usefulness

The first part of the results will be a study of how the skill of Z21 varies according to the different modalities of risk_persistence. To make the link with forecasters' impressions of the risk persistence, the reliability of Z21 probabilities is assessed. It indeed measures the agreement between forecast probabilities and the relative observed frequency of the targeted event (Toth et al., 2003). In practice, it amounts to studying how Z21 may under/overestimate the probability of exceedance depending on whether or not the previous runs also predicted a non-zero probability. By doing this for various precipitation thresholds ranging from 0.2 mm to 100 mm, the dependency of the Z21 reliability on what the previous runs predicted can be highlighted.





### 3.4 Forecast calibration using the risk persistence information

The second part of the results will this time consist of assessing "automatically" the usefulness of risk persistence using a simple machine learning algorithm. The logistic regression is chosen since it has been widely used for precipitation (Ben Bouallègue, 2013, and references therein). If $P(t) = P(RR24 \geqslant t)$ denotes the probability that daily precipitation exceeds "$t$" mm, then the logistic regression derives probabilities through the equation:

$$P(t) = \frac{e^{z(t)}}{1 + e^{z(t)}} \tag{2}$$

where $e^x = exp(x)$ is the exponential function, and $z(t)$ is a linear function of $N$ predictors $X$:

$$z(t) = \beta_0(t) + \sum_{i=1}^{N} \beta_i(t) X_i(t) \tag{3}$$

$\beta_0$ is the regression intercept, whereas $\beta_i$ is the regression coefficient affected to the predictor $X_i$. The only predictors that are considered in this study are raw probabilities, for example from Z21, and risk_persistence seen as a categorical variable. The potential added value of taking into account the risk persistence information is assessed by testing several regressions that differ in the input predictors, mainly in the use or non-use of risk_persistence. Those regressions are compared with each other and with raw probabilities from Z21 and from the lagged ensemble based on the four daily runs (Z03, Z09, Z15, Z21). The regression coefficients that have been estimated for each modality of risk_persistence are also interpreted.

### 3.5 Spatial neighborhood post-processing

In order to cope with the double-penalty effect (Ebert, 2008) and the representativeness issue due to the scale mismatch between forecasts and observations (Ben Bouallègue et al., 2020), a 25-km radius neighborhood is introduced. In concrete terms, the four AROME-EPS runs are used to predict the probability of daily precipitation exceeding a given threshold anywhere within a 25 km radius of the RADOME stations, rather than predicting it precisely at station locations. This is done by using the same upscaling procedure as in Ben Bouallègue and Theis (2013), i.e. the probabilities are generated from the maximum of each member within the area of interest. Observations are also upscaled, taking the maximum of the ANTILOPE QPE algorithm within the area, which includes RADOME stations by construction. Other neighborhood approaches could be used (some can be found in Schwartz and Sobash, 2017), but the upscaling procedure was preferred because of its relevance to the issuance of warnings (Ben Bouallègue and Theis, 2013).

As many RADOME stations are less than 50 km apart, many verification areas will overlap if this neighborhood method is applied to all stations. This is problematic since it can bias the scores estimation and may lead to overfitting (Hastie et al., 2009). To avoid this, only RADOME stations that guarantee non-overlapping verification areas are selected, and from the 596 initial stations, only 164 are finally used. As shown in table 1, the number of daily ⟨obs, forecast⟩ couples is thus much reduced, leading to a total of 59368 over the 362 days of the study period. The upscaling procedure has however the substantial advantage of considering many more (and potentially interesting) precipitation values than only those observed at the rain gauge locations or predicted at specific grid points. In the following, all results are based on a 25-km radius neighborhood





| Neighborhood size | Daily ⟨obs, forecast⟩ couples | Sample size |
|---|---|---|
| 0 km | 596 | 215 752 |
| 25 km | 164 | 59 368 |
| 50 km | 56 | 20 272 |

**Table 1.** Number of daily ⟨obs, forecast⟩ couples and the resulting sample size of the whole study period, for each neighborhood size.

unless explicitly stated otherwise. Their sensitivity to the spatial neighborhood size and the relevance of the 25 km radius will be studied in a dedicated subsection and discussed.

### 3.6 Scores and regression coefficients estimation

Reliability diagrams are used to assess the reliability of forecast probabilities. Here, they are obtained by plotting the relative observed frequency of the exceedance within the following 10 forecast probabilities bins: [0%-10%[, [10%-20%[, ..., [90%-100%]. The statistical significance of the results (including these diagrams) is estimated by a bootstrapping procedure which produces 1000 resampled study periods. Bootstrap mean and Q5-Q95 confidence interval are shown, but only for results whose estimation is sensitive to the sample used.

To complete the methodology, we would like to point out that unlike Ben Bouallègue (2013), there is no unification term in the logistic regression: each exceedance threshold requires a specific estimation of the regression coefficients, which explains why the different elements of equations (2) and (3) depend on "$t$". Regression coefficients are estimated by a 1-day cross-validation procedure with a 5-days block separation between test and training samples. In other words, one day of the study period is selected as test sample, and the regression training is done over all the other days, excluding the 5 days on either side of the test day to ensure complete separation between the test and training samples. At the end of the cross-validation procedure, each day of the study period has been used once as test sample. For further details on logistic regression, the reader could refer to Ben Bouallègue (2013) or Wilks (2011).

## 4 Results

### 4.1 Manual assessment of the risk persistence usefulness

In what follows, we take the perspective of a forecaster confronting the latest AROME-EPS run predicting a risk of a given precipitation threshold being exceeded in 24 hours, i.e. at least one member of Z21 is predicting the exceedance. The objective is to highlight the extent to which he/she has to take the latest run at face value, depending on whether or not the previous runs also predicted a risk. For the sake of clarity, the two "extreme" scenarios of risk persistence are considered: either the risk of exceedance is predicted by Z21 only (i.e. risk_persistence = 0), or it is also predicted by all three previous runs (i.e.



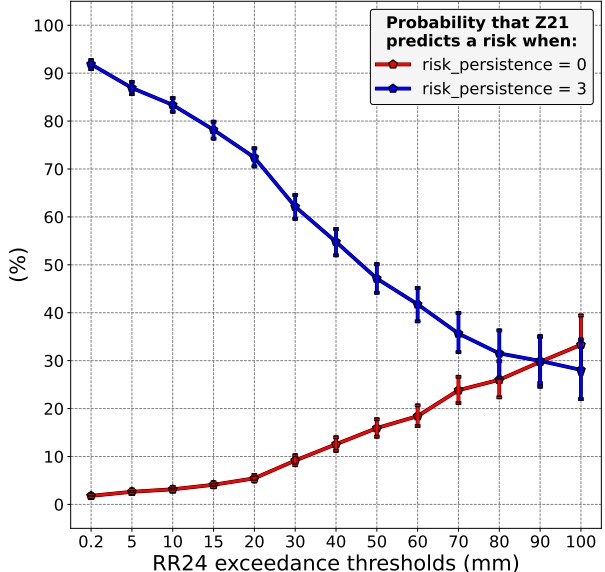

**Figure 3.** For each precipitation amount displayed on the x-axis, the red line represents the probability of Z21 being the only run of the day to predict a non-zero probability of exceedance. Conversely, the blue line shows the probability of Z21 predicting a non-zero probability of exceedance as well as the three previous runs (risk_persistence = 3). Vertical bars indicate 5%-95% confidence intervals.

risk_persistence = 3). Fig 3 first shows what are the chances of a forecaster being in one of those two cases, expressed as a probability in %. The higher the precipitation amount, the rarer the risk_persistence = 3 case (blue line) and the more frequent the risk_persistence = 0 case (red line). In other words, exceedance scenarios involving high amounts of precipitation are less likely to persist over consecutive runs. Besides, around 100mm, both scenarios are equally likely, which makes it relevant to

ask whether a forecaster should act differently in these two cases, as discussed in the following.

To assess the practical usefulness of risk persistence, a comparison between the average probability predicted by Z21 and the observed RR24 exceedance frequency is made in the first place. This simple diagnostic is used to identify possible trivial under/overestimation biases, before conducting any in-depth analysis. This comparison made over the whole study period leads to the two green lines of fig 4. The solid and dashed green lines, refering to the average Z21 probability and the observed

exceedance frequency respectively, are close whatever the precipitation threshold. It means that, on average, Z21 probabilities are a reliable estimate of the risk of exceedance. However, this is not true if this comparison is restricted to the subsample in which Z21 is the only run to predict a risk (risk_persistence = 0), as depicted by the red lines. In this particular case, the solid line is above the dashed line for most precipitation thresholds, which means that Z21 tends to overestimate the probability of these thresholds being exceeded. On the contrary, in the subsample where a non-zero probability is predicted by all the four

daily runs (risk_persistence = 3), the latest run seems to underestimate the probability at most precipitation thresholds (blue lines).





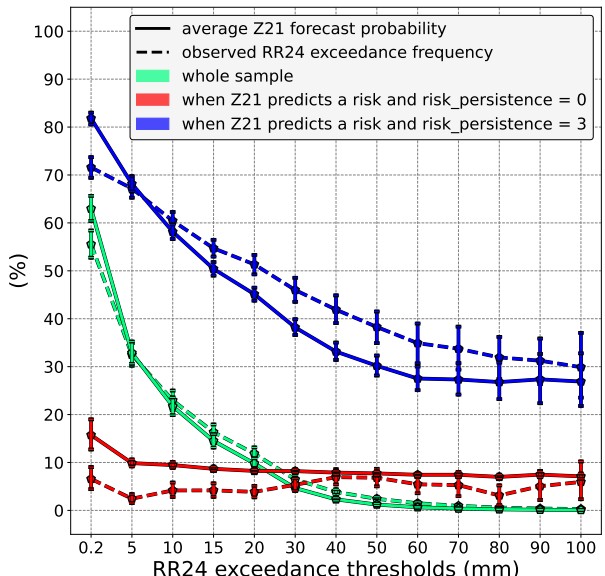

**Figure 4.** Comparison between the average probability of exceedance predicted by Z21 (solid lines) and the observed exceedance frequency (dashed lines), over three different samples. In green, the whole sample is considered. In red, the comparison is restricted to the case when Z21 predicts a non-zero probability but not the 3 previous runs (risk_persistence = 0), whereas in blue, it is predicted by Z21 as well as the 3 previous runs (risk_persistence = 3). Vertical bars indicate 5%-95% confidence intervals.

A further analysis is yet needed to ensure that the risk persistence explains these biases. Indeed, because the average probability predicted by Z21 is significantly higher when risk_persistence = 3 than when risk_persistence = 0, the previous results could just reflect the fact that high probabilities tend to underestimate the real risk of exceedance, whereas low probabilities tend to overestimate it. An appropriate way of testing this hypothesis is to compute a reliability diagram. Fig 5 shows one related to the 20 mm precipitation threshold, for which the under and overestimations made by Z21 are both statistically significant, cf fig 4. Focusing on the risk_persistence = 3 case, fig 5 confirms that Z21 is underestimating the risk of exceedance, but only for probability levels under 60%. For higher precipitation thresholds (not shown), this limit is decreased, i.e. only low levels of probabilities are consistently underestimated. As for the risk_persistence = 0 case, Z21 probabilities are overestimated, but unlike for the risk_persistence = 3 case, Z21 mostly predicts small probabilities as shown by the frequency histogram (shaded bars, cf right y-axis). This remains true for the other precipitation thresholds (not shown). In this respect, it is interesting to remake the comparison shown in fig 4, but restricting it only to the small probabilities predicted by Z21. Fig 6 illustrates this, by focusing the comparison on Z21 probabilities lower than 15%. It reveals that when risk_persistence = 3, the low probabilities predicted by Z21 are largely underestimated at all precipitation thresholds, whereas when risk_persistence = 0, they are overestimated, but to a lesser extent and not for all precipitation thresholds. We also note that the underestimation bias when risk_persistence = 3 is much more obvious than in fig 4, especially for high precipitation amounts.



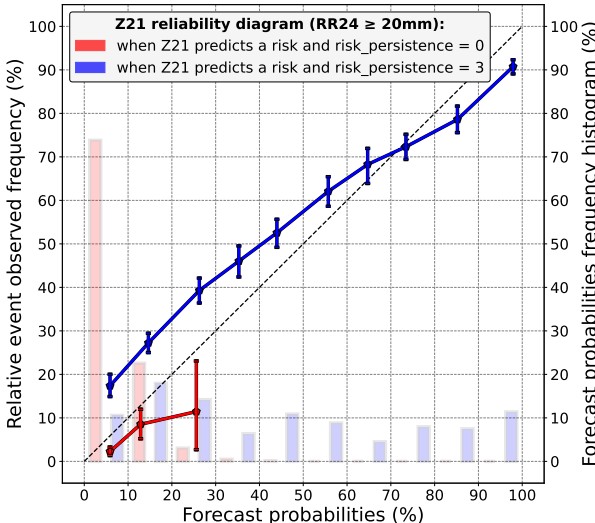

**Figure 5.** Z21 reliability diagram for the 20mm exceedance threshold. In red, the sample is restricted to the case when Z21 predicts a risk but not the three previous runs (risk_persistence = 0), whereas in blue, to the case when a non-zero probability is predicted by Z21 as well as the three previous runs (risk_persistence = 3). For each case, the frequency histogram of Z21 forecast probabilities is also represented by shaded bars (right y-axis). Vertical bars indicate 5%-95% confidence intervals.

Some practical recommendations can be made from these findings. Forecasters should not take the precipitation probabilities of the latest run at face value, without checking in the previous runs whether that exceedance scenario has just emerged or has recurred run after run (possibly in a minority compared to the other members). The probability of this scenario occurring may
be overestimated by the latest run in the former case, and underestimated in the latter case. In particular, forecasters should pay attention to scenarios that have a small non-zero probability according to the latest run, but which have been recurrently forecast by the previous runs. In that case, their likelihood is probably (largely) underestimated by the latest run. Finally, it should be remembered that the two extreme scenarios of risk persistence are about equally likely to occur when it comes to forecasting heavy precipitation (around 100mm, cf fig 3). As forecasters should act differently for these two scenarios, they
should be particularly vigilant for such events.

These recommendations are valid for forecasters who consider runs separately. Thus, a related question is: should a forecaster working with a lagged ensemble ignore such recommendations, or in other words does the contribution of each run to the final lagged ensemble probability matters? By following the same procedure, but using probabilities computed by merging the four daily ensemble runs instead of just Z21, the answer to the last question is also yes, as similar reliability biases depending on
the different risk_persistence modalities were found. As a proof, the next subsection will show how the lagged ensemble, as well as Z21, can benefit from these results by improving their reliability thanks to the risk persistence information.

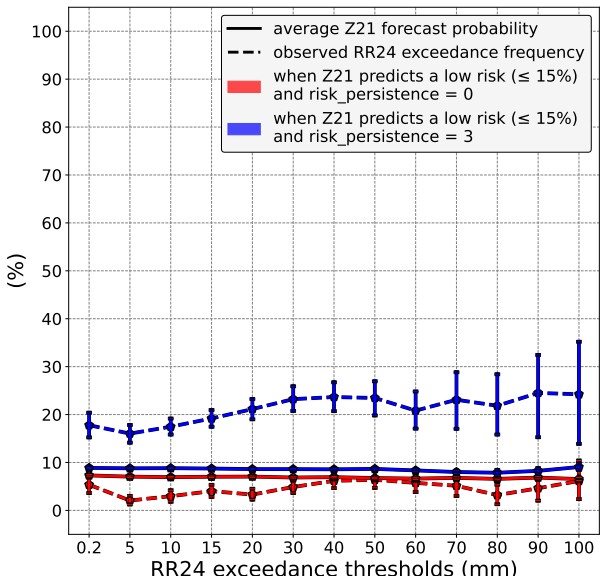

**Figure 6.** Comparison between the average probability of exceedance predicted by Z21 (solid lines) and the observed exceedance frequency (dashed lines). Unlike in fig 4, only the probabilities ≤ 15% predicted by Z21 are considered. In red, the comparison is restricted to the case when Z21 predicts a non-zero probability but not the three previous runs (risk_persistence = 0), whereas in blue, it is predicted by Z21 as well as the three previous runs (risk_persistence = 3). Vertical bars indicate 5%-95% confidence intervals.

## 4.2 Forecast calibration using the risk persistence information

In this section, the usefulness of risk persistence is assessed using a simple machine learning algorithm, the logistic regression. Doing so, we let the algorithm establish a link between the risk of exceedance and the different risk_persistence modalities, and use it to produce more skillful forecasts than raw ensembles. In the following, three regressions are tested. The first regression uses only one input variable, the raw Z21 probabilities. The second regression uses in addition the risk_persistence variable. The third regression works like the second one, but the raw probabilities are taken from the lagged ensemble based on the four daily runs, instead of Z21. These regressions are compared with each other and with raw probabilities from Z21 and the lagged ensemble. As for the previous results, the reliability is assessed.

Fig 7 shows reliability diagrams for two daily precipitation thresholds, 10mm and 30mm. Focusing on the 10mm threshold (left diagram), the latest run and the lagged ensemble (green and purple lines, respectively) are found to be quite reliable, although there is a slight tendency towards probability overestimation. The lagged ensemble is not much better than Z21, and is even worse at the 60%-70% levels. The regression with raw Z21 probabilities as only input variable performs poorly, it even degrades the reliability of the raw ensemble (dark green line). On the contrary, reliability is improved with the risk_persistence information (light blue line). Similar result are obtained with the lagged ensemble (orange line), which demonstrates the usefulness of including the risk persistence information even in the lagging approach. Focusing now on the 30mm precipitation



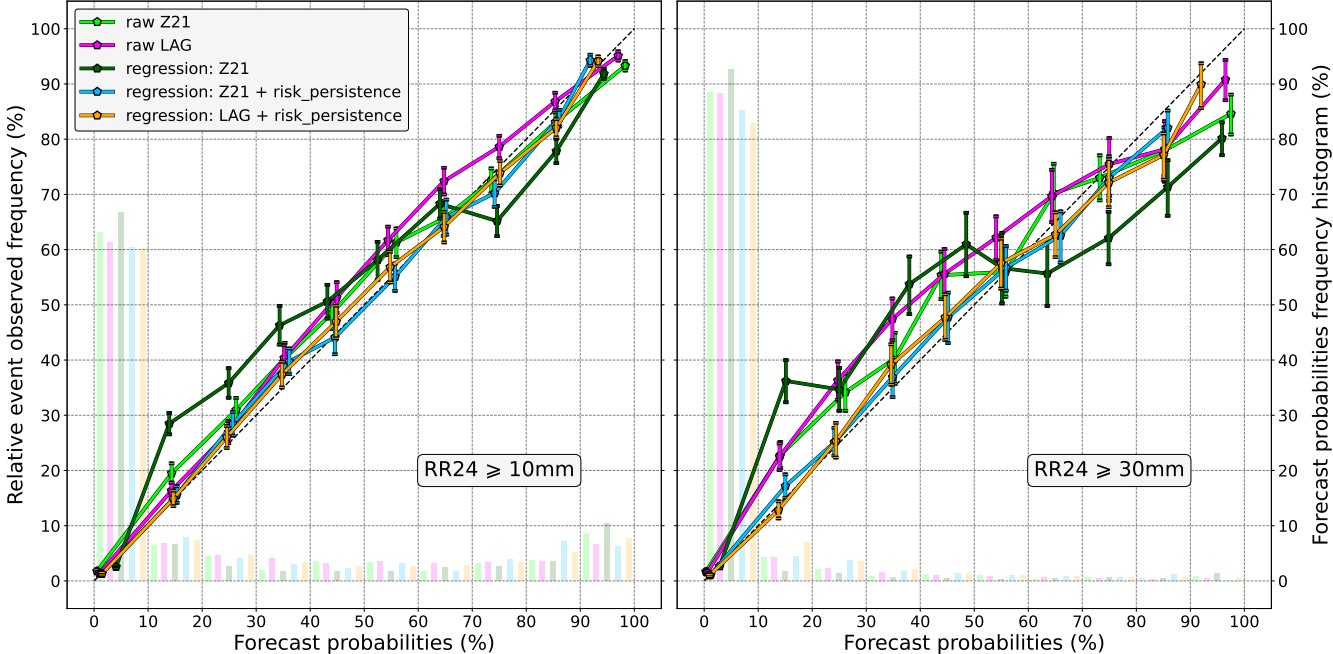

**Figure 7.** Reliability diagrams for the 10mm (left) and 30mm (right) exceedance thresholds. The raw Z21 ensemble is in green, whereas the raw lagged ensemble based on the four daily runs is in purple. The logistic regression with raw Z21 probabilities as only input variable is in dark green. In light blue, the risk_persistence variable is added to that regression. In orange, it is the same regression as in light blue, except that the raw probabilities are derived from the lagged ensemble, instead of Z21. The frequency histogram of the corresponding forecast probabilities is also represented by shaded bars (right y-axis). Vertical bars indicate 5%-95% confidence intervals.

threshold (right diagram), raw ensembles are less reliable. Nonetheless, the regressions that use the risk persistence information still improve the reliability. As for the 10mm threshold, the regression using raw probabilities as the only predictor performs poorly. Other precipitation thresholds have been tested, with similar results, although it has to be noticed that the higher the

threshold, the more difficult the assessment, especially for high levels of probability due to lack of data in the related bins.

To understand how reliability has been improved by these simple regressions, it is interesting to visualize how they transform the input raw probabilities given each risk_persistence modality. Fig 8 shows this dependency for the "blue" regression, and for the 30mm threshold for which the impact of the regression is more obvious. In this figure, each line depicts how the input Z21 raw probabilities (x-axis) are transformed by the logistic regression (y-axis) within a given risk_persistence case. For example,

if risk_persistence = 3 (dark blue line), a probability of 20% predicted by Z21 becomes a 30% probability after the regression. These lines only differ by the specific coefficient affected by the regression to each risk_persistence modality.

When risk_persistence = 3, the probabilities forecast by the latest run are increased by the regression, as the dark blue line is above the diagonal for most probability levels. Also, this upward adjustment of raw probabilities is more important for low levels, even if the difference is small. These results are similar to those shown in the previous subsection, although




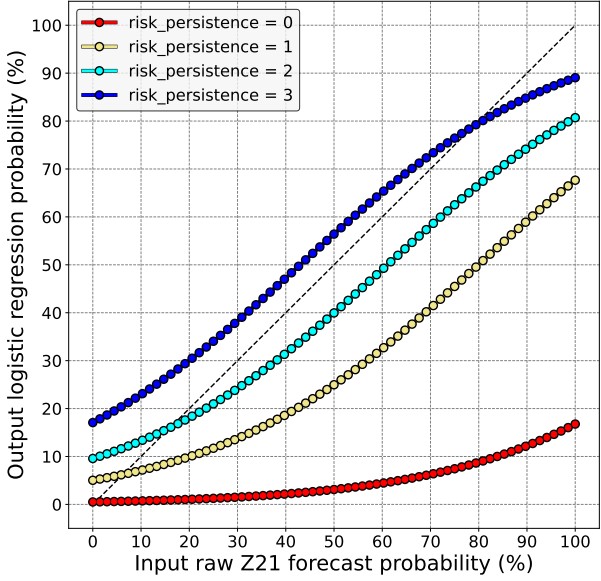

**Figure 8.** In the y-axis, post-processed probabilities of 30mm being exceeded in 24 hours, derived from the logistic regression with Z21 raw probabilities and risk_persistence as only input predictors. They are expressed as a function of Z21 raw probabilities (x-axis), distinguishing each risk_persistence modality (colors).

obtained differently. Raw probabilities are adjusted downwards at all the other risk_persistence modalities, the magnitude of this decrease being roughly proportional to the number of previous runs that did not predict a risk. A spectacular decrease is obtained for the risk_persistence = 0 case (red line), for which the regression almost nullifies the latest run, whatever the probability it predicts. The way raw probabilities are calibrated remains quite the same whatever the precipitation threshold, with only the distance from the diagonal changing slightly (not shown). Fairly comparable results are obtained for the regression with the lagged ensemble, which indicates that probabilities have been similarly calibrated using similarly the risk persistence information, whether derived from the last run or from the lagged ensemble.

### 4.3 Sensitivity of the results to the spatial neighborhood post-processing

As mentioned in the methodology section, all the previous results were obtained with a 25-km radius neighborhood. What would the results be without any spatial neighborhood, or with a larger one, are important questions regarding the effective usefulness of this paper. This has been studied by reproducing the results with a 50-km radius neighborhood and with no spatial neighborhood at all. In the following, only the impact on the logistic regression is shown because it is a direct assessment of the usefulness of the risk persistence information.

In fig 9 are displayed the results obtained without any spatial neighborhood. Focusing on the 10mm threshold (left diagram), what differs from the 25-km radius neighborhood experiment is that raw ensembles are biased in the other sense: raw probabilities are, in average, overestimating the risk of exceedance. Nevertheless, reliability is still improved by using the risk





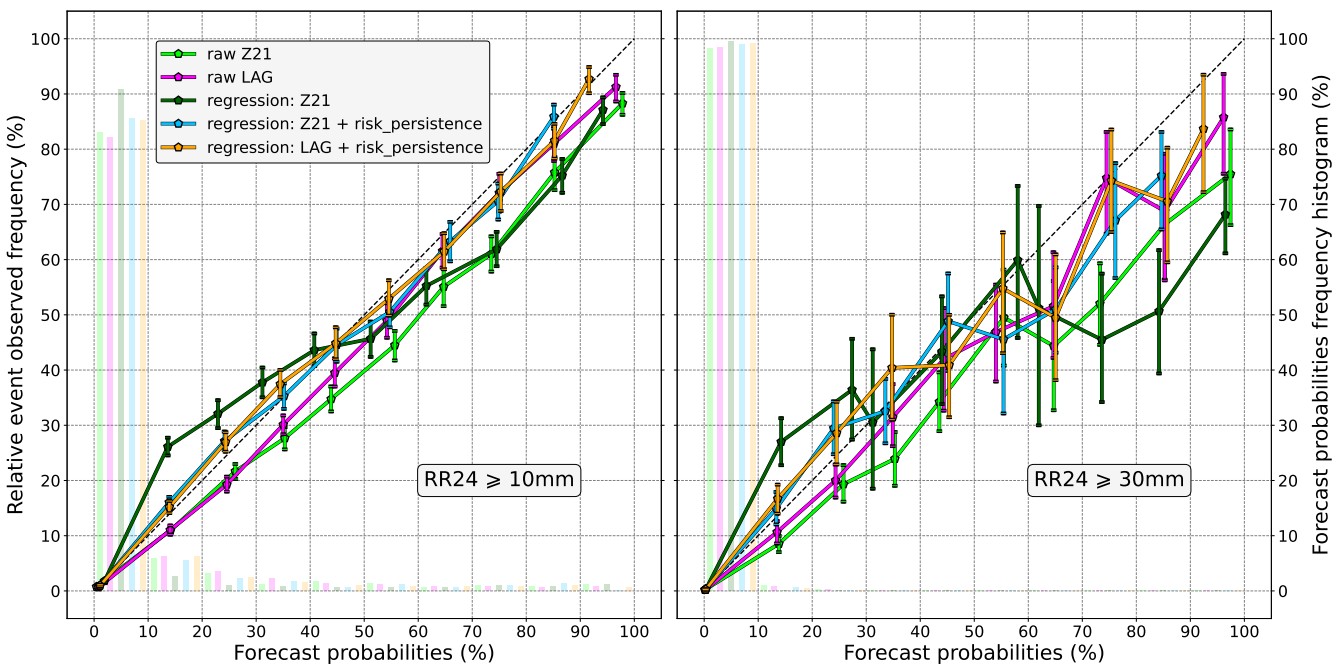

**Figure 9.** As in fig 7, but without any spatial neighborhood.

persistence information, and the regression that does not use it still performs poorly. For higher precipitation thresholds, results are difficult to interpret because of the large confidence intervals and the jumpy lines, as it can be seen in the right diagram (30mm). This problem does not come from the sample size, which is enlarged compared with the 25-km neighborhood as shown in table 1. Indeed, the larger the neighborhood, the lower the sample size, because less verification areas are needed to

cover mainland France without any overlap between them. It rather means that there is not enough data in each probability bin, as shown by the frequency histogram. Very few non-zero probabilities are forecasts, suggesting that it is rare that several AROME-EPS members predict more than 30mm in 24 hours at the exact same grid point.

The results obtained with a 50-km radius neighborhood are shown in fig 10. As an illustration, the spatial coverage of such area is shown in fig 1, centered on the Toulouse-Blagnac station (B), and can be compared to a 25-km radius area centered on

the Biarritz station (A). Because high precipitation thresholds are likely to benefit from upscaling (Ben Bouallègue and Theis, 2013), the 30mm and 50mm exceedance thresholds are assessed. For the 30mm threshold (left diagram), the reliability bias of the raw probabilities is similar but slightly more exaggerated than in the 25-km experiment. Again, this bias can be reduced by using the risk persistence information, in particular for the probabilities up to 30-40%. Similar results are found for higher thresholds, such as 50mm (right diagram), even if they are only valid for levels up to 50% due to lack of data above.



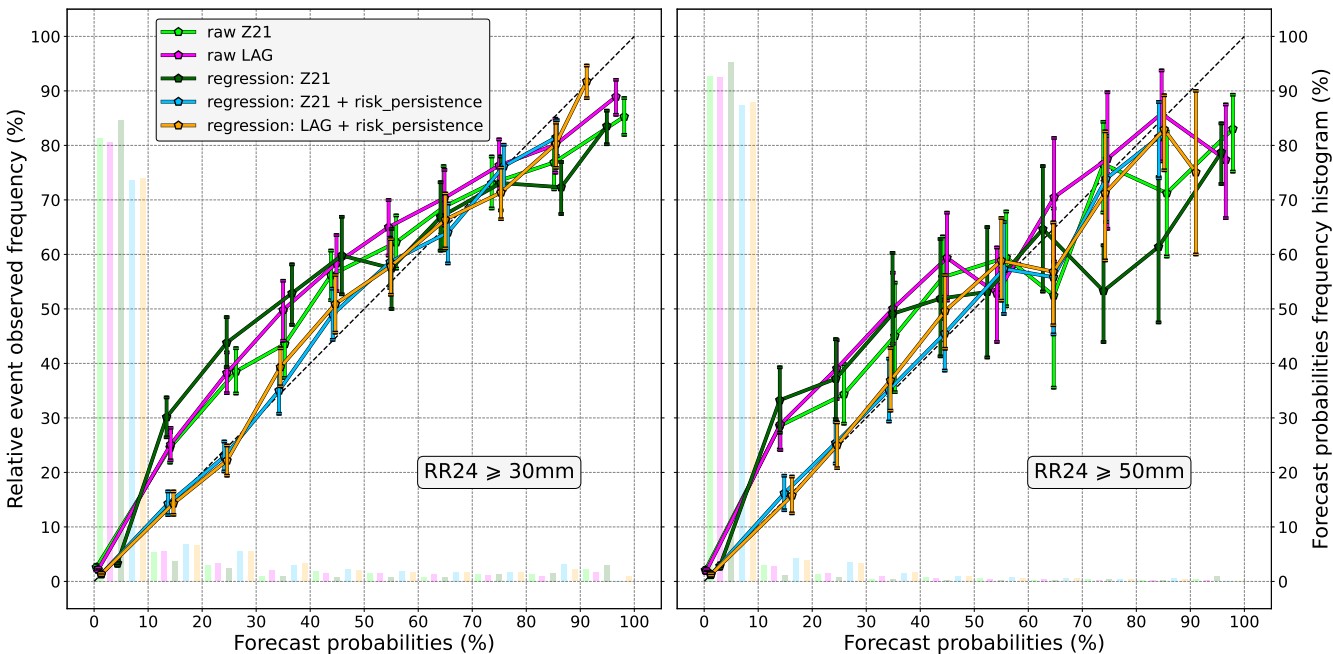

**Figure 10.** As in fig 7, but with a 50-km radius spatial neighborhood, and for the 30mm (left) and 50mm (right) precipitation thresholds.

## 5 Discussion

Several details of this study need to be discussed. To begin with, the risk_persistence definition, on which the entire study is based, does not fully take into account the chronological order of runs. A more refined definition could be:

$$\text{risk\_persistence} = \begin{cases} 0 & \text{if none of Z03, Z09 and Z15 had predicted a risk} \\ 1 & \text{if among Z03, Z09 and Z15, only Z15 had predicted a risk} \\ 2 & \text{if among Z03, Z09 and Z15, only Z15 and Z09 had predicted a risk} \\ 3 & \text{if Z03, Z09 and Z15 had all predicted a risk} \end{cases} \quad (4)$$

Compared to the first definition given by equation 1, the modalities 1 and 2 are more restrictive, so as to better characterize the persistence of a given weather scenario, from brand new (modality 0) to "long-lived" (modality 3). Although appealing, this definition suffers from drawbacks that have led us to reject it. The main problem comes from its difficult use as an input variable to a regression, because many cases of the risk persistence are not covered, such as "among Z03, Z09, Z15, only Z09 had predicted a risk". This is problematic within the regression framework, since the impact of all these missed cases on the risk of exceedance would be blended in one single coefficient, the intercept, making the usefulness of accounting for the risk persistence more difficult to assess. One solution could be to create as many modalities as there are cases, but each case would represent a small subset, leading inevitably to sample size issues that are harmful to the robustness of the results. In the light of





all this, the definition of risk persistence used throughout this study seems to be a good compromise, as it remains quite simple while still providing an information about the chronological order of the runs.

The criterion "at least one member predicts the exceedance" seems a bit ad-hoc and could also be refined. Indeed, the exact
number of members simulating it is not accounted for, although it could be a valuable information. We decided to ignore it, because it would be too complex for a first study, as it would imply dealing with the many possibilities for the evolution of probabilities over consecutive runs, which could not be summarized by a simple 4-modalities variable. As an illustration, an overview of the difficulties to characterize the "trend" feature, i.e. probabilities that increase or decrease run after run, can be found in McLay (2011). It is also worth saying that, beyond the sample size issues this may raise, the definition of such features
would be based on probability thresholding, which would need to be adapted to the different precipitation exceedance thresholds. For instance, the "sneak" and "phantom" sequences described in McLay (2011), which are respectively characterized by large and rapid increases or decreases in event probability at short lag times, can not be used for high precipitation thresholds in their original definition, since high probabilities of exceedance are almost never forecast at such thresholds. Finally, it should not be forgotten that the entire study is based on AROME-EPS, which only comprises 17 members. Given this ensemble size,
the use of quantiles and probabilities can be limited (Leutbecher, 2018). In that context, any non-zero probability is already a strong signal in itself, as already noticed by Mittermaier (2007).

Another aspect to discuss is the use of a spatial neighborhood. In this study, it amounts to introducing a spatial tolerance when forecasts and observations are being compared, as the objective is to predict the probability of daily precipitation exceeding a given threshold anywhere within a given area, rather than predicting it at precise locations. Despite this loss of resolution, the
use of a spatial neighborhood for precipitation has proved relevant in our opinion. For instance, it was found that the prediction of moderate to high precipitation amounts being exceeded could benefit from the risk persistence information only in the event of it. Changing the spatial scale also changes the way different aspects of the forecasts are perceived, which is important to keep in mind. The difference between the frequency histograms of forecast probabilities computed with/without spatial neighborhood illustrates this. Because members rarely agree on the exact location of such precipitation amounts, the risk of
exceeding them is mostly low without spatial neighborhood, and therefore it could appear almost negligible. This impression can be misleading, because that same risk is revised upwards as soon as a spatial neighborhood is introduced, showing that a consensus may appear between the same members within a slightly larger scale. Regarding the neighborhood size itself, the 25-km radius was chosen because it seemed a good compromise. It is wide enough to benefit from the previous advantages, while remaining reasonable, especially for Météo-France forecasters who issue warnings on the department scale.

If this study focuses on reliability, what about other forecast attributes? In particular, the discrimination, i.e. the ability to discriminate between event and non-event, is also an important aspect of probabilistic forecast (Murphy, 1991). This attribute was investigated using ROC curves (Mason, 1982). Surprisingly, the discrimination was neither improved nor degraded by incorporating the risk persistence information. In our opinion, it could be a consequence of the specific way raw probabilities are transformed by the regression. Indeed, an increasing monotonic transformation of probabilities can not modify the
ROC curve as shown in Jolliffe and Stephenson (2012). Some tests have been carried out to verify this hypothesis, in which interaction terms have been added to the regression: they showed no statistically significant added value in both reliability





and discrimination. Understanding why the discrimination is not impacted would be a good step forward, as it would better characterize the effective usefulness of the risk persistence information. But it would require further tests, such as using a more advanced machine learning algorithm, which is out of the scope of this paper. Anyway, we believe that the (high) sensitivity of
the reliability to the risk persistence is already a significant result in itself.

Finally, regarding the results themselves, it has to be underlined that they are somewhat at odds with the current state of the art, since the evolution of forecasts has not previously been reported to be strongly related to the upcoming weather. In our opinion, there could be several reasons for this. First, we tried to maximize the chances to obtain new results on that subject. For example, the present study differs from the previous ones by focusing on a parameter that is well-known for its large spatial and
temporal variability, and on a more tangible aspect of forecasts evolution than an "all-in-one" run-to-run variability measure. We also felt that working with an ensemble was preferable for the question we wanted to explore. Indeed, the idea behind this work is to find out whether the way the atmosphere is evolving, as perceived by a given NWP system, gives us information about the upcoming weather. For this to have any chance of working, the run-to-run variability has to be an accurate reflection of what is happening in the atmosphere. The problem with deterministic models is that they are non-linear systems which are
highly sensitive to small perturbations (Leutbecher and Palmer, 2008): therefore, the variations from one run to another, which are assumed to be strictly caused by the assimilation of new observations, may also be insidiously affected by such sensitivity. Ensembles are less subject to this sensitivity by construction, and are found more consistent from one run to another (Buizza, 2008; Zsoter et al., 2009; Richardson et al., 2020).

Another insight can be found in Richardson et al. (2024). In this article, the origin of run-to-run variability is studied, and
several factors that can influence it are identified and discussed. The data assimilation (DA) algorithm is one of them, and it is clear that changing it would probably affect our results. Independently of this paper, Météo-France is currently testing a three-dimensional ensemble-variational (3DEnVar) DA algorithm (Michel and Brousseau, 2021) for its regional deterministic model AROME-France, and preliminary results show a better consistency between runs, in particular for case studies involving high precipitation, compared to the current three-dimensional variational (3DVar) scheme. The ensemble spread, size and
perturbations are also important factors identified by Richardson et al. (2024). Would the same results as ours be obtained with another ensemble, or with a different number of members, is an interesting and open question. With a bigger ensemble, the probability of having a weather scenario recurring run after run would certainly be higher, as each run would explore a wider range of possibilities. In this context, the risk_persistence = 3 case would occur more often, and the risk persistence as defined in this study may be of limited use. On the basis of this reasoning, our results can be understood differently. Indeed,
it can be hypothesized that if the risk persistence has worked for this study, it is precisely because the information provided by it has somehow compensated for the limited size of AROME-EPS, by (for instance) giving back importance to scenarios whose likelihood was under-represented for "wrong" reasons, e.g. not enough members, etc. If that were true, this study would suggest that limited sized ensembles may suffer from some kind of "memory loss", which can be partly "healed" by providing a record of their previous runs, but in a more subtle way than within the standard lagging approach. And it would finally be
in line with the state of the art, as it would mean that it was not the forecasts evolution that gave us an information on the upcoming weather.



# 6   Conclusions

This paper addresses the issue of forecasting the weather using consecutive runs of one given NWP system. As forecasts may vary (sometimes significantly) from one run to another, this situation can be difficult to deal with. In the literature, considering
how forecasts evolve from one run to another has never been proved relevant to predict the upcoming weather. Therefore the usual approaches to handle this are, either considering only the latest run, or blending all together the successive runs to create a lagged ensemble. However, both approaches suffer from shortcomings, and if the relationship between changes in forecasts and predictability is assumed to be weak, some aspects remain unexplored. This article is an attempt to further assess this relationship.

As forecast evolution can be described in many different ways, we have focused on a simple, tangible aspect of ensembles: the persistence of a given scenario over consecutive runs. Following discussions with Météo-France forecasters, we have investigated the idea that, the more a scenario recurs run after run, the more it is likely to occur, but its likelihood is not necessarily estimated as it should be by the latest run alone. Using the regional ensemble of Météo-France, AROME-EPS, and forecasting the probability of certain (warning) precipitation amounts being exceeded in 24 hours, the notion of "risk
persistence" has been introduced. It characterizes the newness of the weather scenario involving the exceedance, from "long-lived" (it was predicted several runs ago and has recurred run after run) to brand-new (it has just emerged from the latest run).

Doing so, it has been found out that reliability, an important probabilistic forecast attribute, is quite dependent on the risk persistence. In particular, it has been highlighted that the probability predicted by a given run can be under/overestimated,
depending on whether or not the previous runs also predicted a non-zero probability. Similar biases were found for the standard lagged ensemble, suggesting that the contribution of each run to the final lagged probability matters. The usefulness of the risk persistence information has also been assessed using a simple machine learning algorithm, the logistic regression. It has been shown that forecast reliability can be improved, including for moderate to high precipitation amounts, just by providing it.

In our opinion, what should be remembered about this work is not so much the reliability improvement obtained by the
use of a logistic regression. More advanced machine learning algorithms (for instance) would certainly lead to better results. It is rather the high sensitivity of the reliability to something as simple as the risk persistence. This could be seen as a proof that considering forecasts evolution can actually be useful for weather forecasting. But it could also reveal that limited sized ensembles may suffer from some kind of memory loss, as they do not reliably estimate the likelihood of weather scenarios that were recurrently suggested by previous runs. Further studies yet are needed to better understand this point.

At this stage, we see two applications for this study. It could pave the way for the use of new predictors for statistical post-processing, based on consecutive runs. Indeed, we believe that we can benefit from looking at successive runs in ways other than by lagging, and that this study is one example among others. It should also discourage operational forecasters to take raw probabilities at face value, without considering their evolution run after run. Regarding this, it would be very interesting to see if some specific weather phenomena could benefit more from this information than others. For instance, it can be supposed that





phenomena well-known for their low predictability, such as Mediterranean heavy precipitation (Khodayar et al., 2021), should
be much more feared than more "common" ones if they persist run after run.

*Code and data availability.*  The figures of this article and the regression fitting can be reproduced by a code and a dataset that will be made
available after the reviewing process.

*Author contributions.*  HM: conceptualization, data curation, formal analysis, investigation, methodology, software, writing - original draft,
review and editing. FB: investigation, methodology, writing – review and editing. ON: investigation, writing – review and editing.

*Competing interests.*  The contact author has declared that neither of the authors has any competing interests.



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
