# Peer review of "Is considering runs (in)consistency so useless for weather forecasting?"

_Natural Hazards and Earth System Sciences, 2024_

## Author Response (AR1)

**Reviewer n°1**

1. Section 4.1, L206-207. "at least one member of Z21 is predicting the exceedance".
Does this just apply to results in this section (figs 3-6)? Then in 4.2 (regression) you use all cases including where no members of Z21 predict exceedance of a given threshold?

Exactly.

In Section 4.1, from the entire study period defined in 2.3, only the times when Z21 predicts a non-zero probability of exceedance are retained. Then, from this sample, two sub-samples are being compared in figures 3, 4, 5, 6: the one where risk_persistence = 0 (red color), and the one where risk_persistence = 3 (blue color). However, there is an exception, the two green lines in figure 4, which refer to the entire study period, regardless of the value of risk persistence or the value of the probability predicted by Z21. This has been made more explicit in the manuscript (lines 206-207, 219-220, 222-223, 225-226) and in the legends and captions of the corresponding figures (3, 4, 5, 6).

On the contrary, in Sections 4.2 and 4.3, the entire study period is used, which has been made more explicit in line 266.

2. L65-66. "scale mismatch". What is the spatial scale of the ANTILOPE observations (are these gridded?)? How different is this from the AROME-EPS grid scale? NB. the "upscaling" used here will not address any scale mismatch – you are still comparing a model grid box value against an observed (gridded or point) value when you take the maximum over a neighbourhood so any difference due to different scales in model and observations will still apply. But I agree this is a good neighbourhood procedure to address the double penalty issue.

Your question about ANTILOPE is perfectly relevant and I apologise for not including this information in the article. In this study, ANTILOPE provides observations on a regular 0.025 x 0.025 degree latitude-longitude grid (so same resolution as AROME-EPS), but over a sub-domain of AROME-EPS. This information has been added line 121.

Since ANTILOPE and AROME-EPS have the same horizontal resolution, and since their grid points are located on the same latitude/longitude "canvas", there is no longer scale mismatch between them. However, it still exists in the 0-km experiment, when AROME-EPS gridded forecasts are compared to RADOME point observations. The "representativeness error" part has therefore been removed from Section 3.5, and added in Section 4.3, lines 313-315. Thanks for the correction.

3. "overestimation" should be "underestimation"

I guess that this point concerns line 267. If that is correct, you are right, I've reversed the terms. Since I've written these words so many times, it had to happen at least once! Correction done in the manuscript.

4. Fig 7. It is interesting that the regression with just the raw probabilities significantly degrades the performance. Any idea why that is?

I was surprised too. My wild guess is that, because zero or (very) low probabilities are predicted most of the time (cf frequency histograms), the regression coefficient corresponding to the raw probability may be optimal for the first probability bin, but less so for the others.
This is also the case in the regression in which the risk persistence variable is added, but this additional information might have an offsetting effect.

5. L340-341. "any non-zero probability is already a strong signal in itself". Fig 8 (red curve for risk persistence =0) seems to contradict this (also figs 4,5), suggesting need to be cautious if non-zero probability in just latest run?

That statement should be interpreted in line with what has previously been written, regarding the fact that notions such as probabilities should be taken with a pinch of salt when dealing with limited size ensembles. It actually suggests that knowing that something might happen (i.e. that it has a non-zero probability of occurrence) is already a strong signal, and knowing the exact value of its probability may be less important in comparison. M. Mittermaier already had this idea in her 2007 article, when she said: "the presence of any non-zero probability suggesting that something might occur was more important than the magnitude". The corresponding part of the manuscript (lines 348-350) has been clarified.

Interpreted in this way, this statement does not seem to me to contradict the figures 4, 5 and 8.

6. L355-365. Very interesting that the resolution is not affected by the regression, and worth further research as you suggest (future work, not for this paper). While a monotonic transformation of probabilities will not affect the ROC, note that the logistic regression can in principle improve the resolution (you have shown in fig 8 that the raw probabilities are affected differently by the different risk persistence values so this is not a simple monotonic transformation)

We were also surprised by the lack of effect on resolution and agree that understanding why is worth further research, especially since resolution is a more important attribute than reliability for heavy rainfall prediction.

7. L389-396. This is a very interesting aspect of the discussion and definitely worth further investigation. I would expect ensemble size to have a significant impact on the results, but then I would have expected that the regression would have a smaller impact on the lagged ensemble, which did not seem to be the case.

Thank you very much for your comment. We agree that this aspect is definitely worth further investigation, and we'd love to see other people looking in this direction.

**Reviewer n°2**

**Major comments:**

1.   When identifying the research gap, the authors often refer to the chronological order (or evolution) of runs (e.g., lines 69, 87, 88). However, while novel, the proposed methodology does not really consider the chronological order of runs, but only how persistent lagged forecasts are, irrespective of their order (see eq. 1). This is also admitted in the discussion (lines 316—338). Therefore, I would suggest stating upfront in the introduction that the proposed methodology aims at improving the way we leverage lagged model forecasts but, at this stage, it does not yet address how to explicitly consider any added information from the chronological order of those runs.

Indeed, the chronological order is not fully accounted for in this article, and even if this point is stated and justified in the discussion, the repetition of this term, especially in the introduction, may be misleading in relation to the rest of the manuscript.

However, even if the successive runs are not sorted in chronological order in the definition of risk persistence (eq. 1), it still contains an information about the temporal dimension of runs, as the forecasts are considered separately on the basis of their initialisation time. That should not be omitted. Moreover, for us, this article is primarily about further exploring the potential usefulness of considering the (in)consistency of runs, and questionning the usual way of dealing with successive runs, i.e. not taking into account their differences and considering them as equally likely members of some ensemble. In that sense, the comparison with the lagged approach is not an end in itself, but rather a means to an end.

It is therefore important to introduce the article in this context, which implies somehow maintaining the idea of taking into account the temporal dimension of the runs, even if it is not by sorting the successive runs in chronological order and extracting some signal from them.

To satisfy your request, we have:

- left line 69 intact, because it just an example, and at this stage it doesn't bode well for the rest of the article.
- removed the whole sequence lines 87-88, because it was not necessary and it was also misleading as you noted.
- modified the beginning of Section 3.2, which is now more in line with the stated aim of the paper (see end of introduction), and no longer talks about considering the chronological order of runs which again was misleading.
- made the same change at the end of the corresponding part of the discussion (line 366 of the modified manuscript).

2.   Fig. 3, 4, and 6: in these three figures, the authors use continuous (solid) lines to indicate what in the caption is referred to as "probability of exceedance predicted by Z21" (in the two cases with risk-persistence=0 and risk-persistence=3). However, from what I understand, what the authors actually show is the frequency of model forecasts exceeding the considered threshold. Using the word "probability" might therefore cause some confusion; for example, it might lead to think that the authors are referring to probabilities calculated using the logistic model given by Eq. 2, although they are actually different concepts, as clearly stated in line 260 ("more skillful forecasts" vs. "raw ensembles").

We agree that the word "probability" is used many times, sometimes refering to different concepts, which can lead to confusion.

In fig 3, we only consider the sample where Z21 predicts a non-zero probability of exceedance, and of this sample, the frequency (in %) of the risk_persistence = 0 case is plotted in red, the risk_persistence = 3 case in blue. In other words, fig 3 compares the frequency of "brand new" risk of exceedance cases to the frequency of "long lived" risk of exceedance cases. This has been clarified in fig 3 (y-axis, legend, caption) and in lines 209-210.

On the contrary, in figures 4 and 6, the solid lines show the average probability forecast by Z21, calculated over different samples to which the colors refer. Green refers to the entire study period, while the red and blue colors refer to the same sub-samples as in fig 3, i.e. cases where Z21 predicts a brand-new risk of exceedance (risk_persistence = 0) and cases where Z21 predicts a long-lived (or persistent) risk of exceedance (risk_persistence = 3), respectively. This has been clarified in figures 4 and 6 (legend, caption) and in lines 216-217.

Hence, whether in fig 3 or in figs 4 and 6, there is never a question of counting the number of forecast probabilities greater than zero, and displaying the corresponding frequency, if that is what you understood.

3.   Fig. 3: in my understanding, another conclusion that can be derived from this plot is that smaller precipitation events tend to be predicted more consistently by all model runs. If this is correct, I would state this in the article too.

This information has been added line 213.

4.   Fig. 3 and 4: what is the difference between the blue and red solid curves shown in Fig. 3 and those shown in Fig. 4? Perhaps, using more informative axis labels for the y-axis may help improve clarity.

I believe that this confusion is caused by the excessive and sometimes inappropriate use of the word "probability", which is the subject of your major comment n°2. The difference between the solid lines of fig 3 and fig 4 has been clarified in response of major comment n°2, and better explained in the text on this occasion.

5.   Lines 172-190 & Fig. 9 and 10: When studying the effects of station sampling, the authors considered the sensitivity to neighborhood size. I suggest also trying different alternative samples of stations for the same, fixed neighborhood size. To validate the proposed methodology, I believe it is more important to observe that results are consistent across different station network realizations with the same 25-km neighborhood size, because of the considerations outlined at lines 181—184.

For the 25-km and 50-km neighborhood experiments, stations are selected in such a way as to avoid overlap between verification areas, while maximising the coverage (i.e. the number of retained stations).

This can be seen as an optimisation problem, which consists of placing as many 25-km (or 50-km) radius discs as possible in metropolitan France, but with the specific condition that the discs can't be placed anywhere, because they have to be centred on a station. This is not a trivial problem to solve, and despite a lot of research, we haven't found a "turnkey" method that guarantees to find the optimal configuration. We have therefore developed a straightforward algorithm which empirically finds one by testing a very large number of different configurations.

The resulting best configurations, i.e. those that retain the most stations, are surprisingly very similar. They always lead to the same stations, with the exception of a few (typically less than three) that are mainly located in geographical areas such as capes, or in areas with a high density of stations, i.e. where the location of one verification area has little effect on the location of the others. This can be seen in the verification areas used for the 25-km and 50-km neighborhood experiments, that we have added in the appendix for you information. As you can see, there is room to add new verification areas, but in practice this is not possible because they have to be centred on a station. Under this constraint very few verification zones can be moved, and overall there is relatively little room for manoeuvre. In other words, if we want to keep as many stations as possible, there are very few possible different samples, probably because some stations are (much) better at optimising coverage than others.

In the light of these considerations, we believe that assessing the sensibility of the results to the station sampling is not necessary, or at least in the way it is proposed. In a slightly different vein, several aspects could be studied, such as the sensitivity of the results to the geographical area, the density of the verification zones (for example, by no longer requiring verification areas to be centred on a station), or the shape of the neighborhood (for instance, squares rather than disks), but these could be an article in themselves and we would be deviating from the main purpose of the article.

To take into account this major comment, the previous considerations have been added in the discussion (lines 363-369), and the title of Section 4.3 has been modified. Also, the verification areas for the 25-km and 50-km neighborhood experiments have been added in the Appendix.

6. The proposed risk-persistence metric is validated by comparing forecasted probabilities with the observed frequencies, for fixed precipitation threshold exceedance (e.g., Fig. 5 and 7), as well as the forecasted and observed frequencies for varying thresholds (e.g., Fig 4). To obtain a more intuitive assessment of the importance of considering persistent forecasts in lagged model runs, I suggest also considering an event-by-event analysis, counting the number of times persistent (or brand new) forecasts made on day D are consistent with the amount of precipitation that is actually observed on day D+1 (maybe considering some tolerance), as well as the number of times those forecasts over- or under-estimated the actual precipitation amount.

As I understand your comment, what you are suggesting is to assess the ability of persistent (or brand-new) forecasts to predict the amount of precipitation, rather than just the exceedance of some thresholds. Unfortunately, this interesting idea cannot be applied as such in the article. As you noted in the major comment n°1, the entire study is based on the notion of risk persistence, which is used to discriminate forecasts that predict a "brand-new" risk of exceedance from forecasts that are consistent with previous runs, in terms of the possible occurrence of exceedance. Thus, in this paper, your concepts of persistent/brand-new forecasts (these terms never really appear as such in the article) only make sense for a given exceedance threshold, and consequently, a given forecast may be persistent for one threshold, and brand-new for another. This means that the assessment you propose must be done separately, for fixed exceedance thresholds. From here, I am not quite sure to understand how forecasts could be assessed on predicting the precipitation amount. To me there are two possibilities:

- After distinguishing persistent and brand-new forecasts on the basis of a given exceedance threshold, let's say 30mm, we could directly assess their ability to predict the exact amount of precipitation, whatever it is. But this seems contradictory to us, because persistent and brand-new forecasts are assessed on something that has no connection with what make them different, i.e. quantitative precipitation, whether heavy or light, whether related to that threshold or not. Also, since forecasts are persistent/brand-new only on the basis of a given exceedance threshold, the persistent/brand-new nature of a forecast is quite versatile, and the results we would find would have more to do with the threshold, rather than the persistent/brand-new nature (which is what you seemed to be asking for).
- Otherwise, after distinguishing persistent and brand-new forecasts on the basis of a given exceedance threshold, let's say 30mm, we could study how close both are to this threshold, as well as how many times both over- or under-estimate it. But this does not make much sense either, as the "risk" in risk persistence stands for the probability to exceed a given threshold, not to be equal to. The difference between brand-new and persistent forecasts has nothing to do with how close they are to that threshold, so it is a bit strange to assess them on that.

Thus, in order to properly focus the assessment on quantitative precipitation, things need to be done differently. To keep it simple, we could keep the same framework and focus the paper on the probability that the amount of precipitation will be close to, rather than greater or equal than, a given value. In doing so, assessing persistent and brand-new forecasts means looking at how close they are to various thresholds. But this means redoing an entire study, except that there would be an additional degree of freedom: the tolerance threshold used to distinguish forecasts that are close to a given precipitation amount from those that are not. While a study of this type would be interesting, particularly in comparison with the results on thresholds exceedance, we think it would be more relevant to produce two separate papers. This is also motivated by the fact we believe that the framework used in our study is not the most appropriate to focus on quantitative precipitation. Surely a better solution would be to construct the concepts of persistent/brand-new forecasts in a different way, by assessing "statistically" the consistency between runs, i.e. by comparing their pdf, as it has been done in recent work by Richardson et al. (2020, 2024) but for different parameters than accumulated precipitation. This way, brand-new and persistent forecasts could be directly assessed on quantitative precipitation, using statistics-oriented scores like CRPS.

Finally, we would like to point out that although predicting quantitative precipitation is essential, it is also important to focus on the risk of certain precipitation amounts being exceeded, as many decisions are based of threshold exceedance, such as weather warnings (e.g. Ben Bouallègue and Theis, 2013) or flood management (e.g. Pappenberger et al., 2011).

7. Fig. 4: considering different benchmarks (dashed lines) depending on the value of risk persistence is a bit counterintuitive, since the final goal of the work is to determine whether using the proposed metric improves our forecasting capabilities. Probably a better way (also see my previous comment) would be instead showing how often Z21 with risk_persistence=0 and risk_persistence=3, respectively, correctly forecasts next-day precipitation amounts (or threshold exceedances, to be consistent with the definition of risk persistence given by Eq. 1), as well as how many times those forecasts are incorrect instead, on an event-by-event basis. This way, the benchmark is the same for both scenarios (i.e., with risk_persistence=0 and risk_persistence=3) and forecasters can better understand whether considering the persistence of forecasted threshold exceedances in previous runs can help achieve better performances.

I am not quite sure to understand what do you mean by "different benchmarks". Fig 4 compares the average probability predicted by Z21 (solid lines) with the observed exceedance frequency (dashed lines) over different samples, including the risk_persistence = 0 cases (red color) and the risk_persistence=3 cases (blue color). So, in a way, fig 4 already shows the skill of Z21 for both cases, which is what you seem to be suggesting ("showing how often Z21 with risk_persistence=0 and risk_persistence=3, respectively, correctly forecasts next-day precipitation amounts (or threshold exceedances, to be consistent with the definition of risk persistence given by Eq. 1)").

More generally, section 4.1 shows in concrete terms how the quality of Z21 varies according to the value of risk persistence, by focusing on a tangible aspect of probabilistic forecasts (reliability), rather than an "all-in-one" score (CRPS, BS, ...) which mixes several quality attributes. In this way, we believe that forecasters can easily measure the usefulness of risk persistence, because it provides an "a priori" idea of forecast bias (over- or under- estimation of the "real" risk of exceedance).

**Minor comments:**

1.  Line 172: the double-penalty effect is mentioned but not explained; while a literature reference is provided, I would suggest to also include a brief description in the text.

A brief explanation has been added just after.

2.  Line 113: "… as already experienced by many forecasters". Please include one or two references about the observed large variability, both in space and time, of accumulated precipitation, at the end of this sentence.

As requested, three references in which the spatial and temporal variability of observed accumulated precipitation have been added. And to echo the "challenging to predict" part, a recent case study written by forecasters has also been mentionned (Paris 2024 Olympic Games opening ceremony).

3.  Some parts of the manuscript need some rewording to enhance clarity (e.g., lines 151-156)

After the reviewing process, the editor arranges for the manuscript to be proofread by a professional. Be sure that the overall clarity of the manuscript will be improved on this occasion.

4.  Fig 3: for the largest RR24 thresholds, confidence intervals for the two curves overlap and cannot be seen.

The points have been shifted slightly to make it easier to read the confidence intervals.

5.  Fig. 7, 9, and 10 are a bit too "crowded". Can the authors devise better visualization strategies?

It is true that the figures are loaded. Unfortunately, we can't really spread the different lines over several new figures, because we need to be able to compare them all. We have also tried to change the line style, introducing dashed or dotted lines for example, but this reduces the overall clarity. Note that, according to us, the main problem really starts beyond the point where the confidence intervals are "disproportionately" large and the lines start to zigzag. As explained in the manuscript, this indicates a lack of data in the corresponding probability bins and therefore it is not possible to properly interpret the results beyond this point, which makes the poor readability less "harmfull". Up to this point, we believe that the figures are quite readable (even if it means that we sometimes may have to zoom in a bit on the pdf), which is the most important.